# Genome-Wide Mapping Reveals Complex Regulatory Activities of BfmR in *Pseudomonas aeruginosa*

**DOI:** 10.3390/microorganisms9030485

**Published:** 2021-02-25

**Authors:** Ke Fan, Qiao Cao, Lefu Lan

**Affiliations:** 1University of Chinese Academy of Sciences, No. 19A Yuquan Road, Beijing 100049, China; s14-fanke@simm.ac.cn; 2State Key Laboratory of Drug Research, Shanghai Institute of Materia Medica, Chinese Academy of Sciences, Shanghai 201203, China; qiaocao2011@foxmail.com; 3School of Pharmaceutical Science and Technology, Hangzhou Institute for Advanced Study, UCAS, Hangzhou 310024, China; 4NMPA Key Laboratory for Testing Technology of Pharmaceutical Microbiology, Shanghai Institute for Food and Drug Control, Shanghai 201203, China

**Keywords:** *Pseudomonas aeruginosa*, two-component system, BfmRS, regulon, biofilm formation

## Abstract

BfmR is a response regulator that modulates diverse pathogenic phenotypes and induces an acute-to-chronic virulence switch in *Pseudomonas aeruginosa*, an important human pathogen causing serious nosocomial infections. However, the mechanisms of action of BfmR remain largely unknown. Here, using chromatin immunoprecipitation followed by high-throughput sequencing (ChIP-seq), we showed that 174 chromosomal regions of *P. aeruginosa* MPAO1 genome were highly enriched by coimmunoprecipitation with a C-terminal Flag-tagged BfmR. Integration of these data with global transcriptome analyses revealed that 172 genes in 106 predicted transcription units are potential targets for BfmR. We determined that BfmR binds to and modulates the promoter activity of genes encoding transcriptional regulators CzcR, ExsA, and PhoB. Intriguingly, BfmR bound to the promoters of a number of genes belong to either CzcR or PhoB regulon, or both, indicating that CzcRS and PhoBR two-component systems (TCSs) deeply feed into the BfmR-mediated regulatory network. In addition, we demonstrated that *phoB* is required for BfmR to promote the biofilm formation by *P. aeruginosa*. These results delineate the direct BfmR regulon and exemplify the complexity of BfmR-mediated regulation of cellular functions in *P. aeruginosa*.

## 1. Introduction

*Pseudomonas aeruginosa* is a major cause of nosocomial infections worldwide and, in some case, the primary cause of death [1,2]. The World Health Organization has categorized *P. aeruginosa* as a critical priority pathogen and declared an urgent need of new antibiotics [3,4]. Understanding the molecular mechanisms involved in making *P. aeruginosa* a successful pathogen may allow for the development of novel strategies for the prevention and treatment of infections caused by this formidable pathogen [5,6,7,8].

As an opportunistic pathogen, *P. aeruginosa* can thrive in a wide range of habitats [9]. This is achieved, at least in part, by precisely regulating gene expression in response to environmental changes via two-component systems (TCSs), a prevalent signal transduction mechanism in bacteria [10]. *P. aeruginosa* genomes encode one of the largest groups of TCS proteins (i.e., ~130 in PAO1 strain) identified in bacteria, and this is thought to confer exceptional adaptability of this bacterium to a wide range of hosts [7,11,12,13]. TCS typically consists of a membrane-bound histidine kinase (HK) that senses a specific environmental stimulus and a corresponding response regulator (RR) that mediates the cellular response [14,15]. Usually, the response regulator is a DNA binding protein, which contributes directly to the control of gene expression [14,15]. Many TCSs have been described as having a key role during the infection process, whereas only a few have been characterized in great detail in *P. aeruginosa* [13,16].

*P. aeruginosa* BfmRS TCS is comprised of HK BfmS and the RR BfmR (Biofilm maturation Regulator) [7,11,17]. The system got its name from its role in the maturation of biofilm that serves as a hallmark of chronic infections [17]. BfmR is essential for the *P. aeruginosa* biofilm development, most likely via activating the expression of *phdA* that encodes prevent-host-death family protein A (PhdA) [18]. In addition to inducing biofilm formation, BfmR inhibits Rhl quorum-sensing (QS) system and decreases the ability of *P. aeruginosa* to cause infection in lettuce leaf and in animal infection models [7,11]. Intriguingly, BfmR activation was sufficient to direct *P. aeruginosa* toward a chronic infection state (i.e., decreased QS signal production, attenuation of acute virulence, and enhanced biofilm formation) and can also account for adaptive gene expression changes in *P. aeruginosa* DK2 lineage [7], a transmissible clone isolated from chronically infected Danish patients with CF over a period of 38 years [19,20,21]. However, the mechanisms of action of BfmR in *P. aeruginosa* remain poorly understood, and only a handful of BfmR targets (e.g., *bfmRS* operon, *pa4103*-*pa4104* operon, *pa4107*-*pa4106*-*pa4105* operon, *rhlR*, and *phdA*) have been identified so far [11,18].

In this study, we used high-throughput methods to identify BfmR targets. We showed at least 172 genes in 106 predicted transcription units (TUs) are likely under the direct control of BfmR. We provided evidence that BfmR works cooperatively with two-component response regulators CzcR and PhoB, in regulating the transcription of a number of *P. aeruginosa* genes. Our results also highlight a complex role of BfmRS in regulation of biofilm formation and Rhl QS in *P. aeruginosa*.

## 2. Materials and Methods

### 2.1. Bacterial Strains, Plasmids, and Growth Conditions

The bacterial strains and plasmids used in this study are listed in Appendix A. *P. aeruginosa* MPAO1 [22] and its derivatives were grown in Luria–Bertani (LB) broth, Pyocyanin production broth (20 g peptone, 1.4 g MgCl_2_, 10 g K_2_SO4, 20 mL glycerol per liter; pH 7.0) [23], M8-based minimal (MM) medium (6 g Na_2_HPO_4_, 3 g KH_2_PO4, 0.5 g NaCl, 0.2% glucose (*w*/*v*), 0.24 g MgSO_4_, 0.5 g glutamate per liter; pH 7.4) [24], or in low-phosphate M8-based minimal (LPM) medium (45.6 mg K_2_HPO_4_, 13.6 mg KH_2_PO4, 1.6 g NaCl, 0.2% glucose (*w*/*v*), 0.24 g MgSO_4_, 0.5 g glutamate per liter; pH 7.0). *Escherichia coli* cultures were grown in Luria–Bertani (LB) broth. All cultures were incubated at 37 °C with shaking (250 rpm). For plasmid maintenance, antibiotics were used at the following concentrations where appropriate: for *E. coli*, carbenicillin at 100 μg/mL, kanamycin at 50 μg/mL, tetracycline at 10 μg/mL, gentamicin at 10 μg/mL; for *P. aeruginosa*, carbenicillin at 100 μg/mL, and kanamycin at 150 μg/mL, gentamicin at 30 μg/mL in LB and 150 μg/mL in Pseudomonas Isolation Agar (PIA, BD), tetracycline at 30 μg/mL in LB and 150 μg/mL in PIA.

### 2.2. Construction of Vectors

Genomic DNA from *P. aeruginosa* MPAO1 was used as a template for the polymerase chain reactions (PCRs). For generating pAK1900-*phoB* (p-*phoB*), a ~0.8 kb PCR product covering 70 bp of the *phoB* upstream region, the *phoB* gene, and 90 bp downstream of *phoB* was amplified using the primer pair comp-*phoB*-F/R (*Bam*HI/*Kpn*I) (Appendix A). For generating pAK1900-*czcS* (p-*czcS*), a ~1.5 kb PCR product covering 101 bp of *czcS* upstream region and the *czcS* gene of *czcS* was amplified using the primer pair comp-*czcS*-F/R (*Hin*dIII/*Bam*HI) (Appendix A). All the PCR products were digested with corresponding enzymes and cloned into pAK1900 [25] in the same orientation as p*lac*.

For generating mini-*phoB*-*flag* (Appendix A), a ~1.3 kb PCR product covering 561 bp of the *phoB* upstream region, the *phoB* gene (not including the stop codon) was amplified using primers pair mini-*phoB-flag*-F/R (*Hin*dIII/*Bam*HI) (Appendix A), generating *p*-*phoB*-*flag*, and then *p*-*phoB*-*flag* were cloned into integrated vector mini-CTX-*lacZ* [26]. For generating mini-*phoB*-D-*flag* (a mini-*phoB*-*flag* variant with deletion of GACACA in the BfmR-binding site of the *phoB* promoter) (Appendix A), the DNA fragment was amplified using primers mini-*phoB-flag*-F/R (*Hin*dIII/*Bam*HI) (Appendix A) and cloned into pMD19-T by TA-cloning method. Subsequently, primer pair mini-*phoB*-D-*flag*-F/R and a QuikChange II site-directed mutagenesis kit (StrataGene, catalog^#^: 200518) were used to generate *p*-(D)-*phoB*-*flag*, which was further clone into the mini-CTX-*lacZ Hin*dIII-*Bam*HI site.

To construct *czcC*-*lux*, the *czcC* promoter region (−825 to +135 of the start codon of *czcC*) was amplified using the prime pair mini-*czcC*-*lux*-F/R (*Hin*dIII/*Bam*HI). The DNA fragments were further inserted into the mini-CTX-lux plasmid, and the orientation of the *czcC* promoter was in the same direction as the promoterless *luxCDABE*. Similarly, to generate *czcR*-*lux*, a DNA fragment covering the promoter region of *czcR* (−647 to +313 of the start codon of *czcR*) was amplified with prime pair mini-*czcR*-*lux*-F/R (*Hin*dIII/*Bam*HI) and cloned into mini-CTX-lux. For generating *exsA*-*lux*, the *exsA* promoter region (−366 to +26 of the start codon of *exsA*) was amplified using prime pair mini-*exsA*-*lux*-F/R (*Hin*dIII/*Bam*HI) (Appendix A) and cloned into mini-CTX-*lux* (Appendix A). 

All constructs were sequenced to ensure that no unwanted mutations resulted.

### 2.3. Construction of P. aeruginosa ΔczcRS and ΔphoBΔbfmS Mutants

To construct Δ*czcRS* and Δ*phoB*Δ*bfmS* mutants (Appendix A), a SacB-based strategy was employed as previously described [27]. For the construction of the *czcRS* null mutant (Δ*czcRS*), PCRs were performed in order to amplify sequences upstream (~1.1 kb) and downstream (~1.0 kb) of the intended deletion (+60 of *czcR* to +1478 of *czcS*). The upstream fragment was amplified from MPAO1 genomic DNA using primers pair D-*czcRS*-up-F/R (*Eco*RI/*Bam*HI) (Appendix A), while the downstream fragment was amplified with primers pair D-*czcRS*-down-F/R (*Bam*HI/*Hin*dIII) (Appendix A). The two PCR products were digested and then cloned into the *Eco*RI/*Hin*dIII-digested gene replacement vector pEX18Ap, yielding pEX18Ap::*czcRS*UD. Subsequently, a 1.8 kb gentamicin resistance cassette cut from pPS858 (Appendix A) with *Bam*HI was cloned into pEX18Ap::*czcRS*UD, yielding pEX18Ap::*czcRS*UGD (Appendix A). The resultant plasmid, pEX18Ap::*czcRS*UGD, was electroporated into WT MPAO1 with selection for gentamicin resistance. Colonies were screened for resistance to gentamicin and sucrose and sensitivity to carbenicillin, which typically indicates a double-cross-over event, and thus of gene replacement occurring. The gentamicin resistance cassette of the Δ*czcRS*::*Gm* mutant was further excised by using the plasmid pFLP2 that encoded Flp recombinase, yielding the Δ*czcRS* mutant. The deletion of *czcRS* in MPAO1 was further confirmed by PCR.

To construct the Δ*phoB*Δ*bfmS* mutants, firstly *phoB* null mutant (Δ*phoB*) was constructed and a similar strategy as described above was used. PCRs were performed in order to amplify sequences upstream (~1.0 kb) and downstream (~1.0 kb) of the intended deletion (+11 to +700 of *phoB*). The upstream fragment was amplified from MPAO1 genomic DNA using primers pair D-*phoB*-up-F/R (*Eco*RI/*Bam*HI) while the downstream fragment was amplified with primers pair D-*phoB*-down-F/R (*Bam*HI/*Hin*dIII) (Appendix A). The gentamicin resistance cassette of the Δ*phoB*::*Gm* mutant was further excised by using the plasmid pFLP2 that encoded Flp recombinase, yielding the Δ*phoB* mutant. For the construction of Δ*phoB*Δ*bfmS*, the pEX18Ap::*bfmS*UGD plasmid (Appendix A) was electroporated into the Δ*phoB* mutant. The Δ*phoB*Δ*bfmS* mutant was subsequently selected and confirmed by PCR.

### 2.4. Expression and Purifications of Recombinant Proteins CzcR-His_6_ and His_6_-BfmR

Proteins were expressed in *E. coli* strain BL21 star (DE3) and purification was performed as described in previous studies [28,29]. Generally, bacteria were grown at 37 °C overnight in 10 mL of LB medium with shaking (250 rpm). The cultures were transferred into 1 L of LB medium (containing 50 µg/mL kanamycin) incubated at 37 °C with shaking (200 rpm) until the OD_600_ reached 0.6, and then IPTG (isopropyl-1-thio-β-d-galactopyranoside) was added to a final concentration of 1.0 mM. After 20 h incubation at 16 °C with shaking (200 rpm), the cells were harvested by centrifugation and stored at –80 °C.

To purify N-terminal His_6_-tagged BfmR (i.e., His_6_-BfmR), the cells were lysed at 4 °C by sonication in buffer A (10 mM Tris/HCl, pH 8; 150 mM NaCl, 1 mM DTT, 20 mM imidazole). The whole cell fraction was subjected to centrifugation at 4 °C at 12,000 rpm for 25 min to remove insoluble material and the membrane fraction. Clarified cell lysate was loaded onto a HisTrap HP column (GE Healthcare, #17-5247-01), equilibrated with buffer A, and eluted with a 0–100% gradient of buffer B (10 mM Tris/HCl, pH 8; 150 mM NaCl, 1 mM DTT, 400 mM imidazole). The fractions containing recombinant proteins were collected and loaded onto a HisTrap Desalting 5 × 5 mL (Sephadex G-25 S) (GE Healthcare, #17-1408-01) and eluted with a running condition of 10 mM Tris/HCl, pH 8; 150 mM NaCl and 1 mM DTT to remove the imidazole. The purified proteins were >90% pure as estimated by a 10% (*w*/*v*) SDS-PAGE gel followed by Coomassie blue staining.

A similar strategy as described above was used to purify C-terminal His_6_-tagged CzcR (i.e., CzcR-His_6_) expressed from 22b-CzcR-His_6_ (Appendix A). For generating 22b-CzcR-His_6_, the PCR primer pair 22b-*czcR*-F/R (*Nde*I/*Xho*I) (Appendix A) was designed to allow in-frame fusion at the C-terminal end with the His tag from pET22b (Novagen), and the construct was sequenced to ensure that no unwanted mutations resulted.

### 2.5. Chromatin Immunoprecipitation (ChIP)

For generating Δ*bfmRS*::*bfmR*-*flag* strain (Appendix A), we introduced mini-*bfmR*-*flag* (Appendix A) into Δ*bfmRS* mutant (Appendix A), and the construct was intergraded into the *attB* site through a diparental mating using *E. coli* S17 λ-pir as the donor as described previously [7,11]. 

Chromatin immunoprecipitation was modified from existing protocol [30,31]. For BfmR ChIP under LPM-6h condition, we performed the experiments with three biological replicates. Overnight LPM cultures of Δ*bfmRS*::*bfmR*-*flag* were diluted to OD_600_ ≈ 0.05 with fresh LPM medium and grown in a 250 mL Erlenmeyer flask with a flask volume-to-medium volume ratio of 5:1, shaking with 250 rpm at 37 °C. After culturing for 6 h, protein-DNA complexes were cross-linked by addition of formaldehyde (final concentration 1%) and incubated at 37 °C for ten minutes shaking with 250 rpm. Crosslinking was stopped by addition of glycine (final concentration 125 mM). The final OD_600_ was 0.5 and cells were collected from 60 mL culture via centrifugation and washed twice with Tris-buffered saline (20 mM Tris-HCl pH 8.0, 150 mM NaCl). Pellets were re-suspended in 1.5 mL Immunoprecipitation buffer (IP buffer) (50 mM HEPES-KOH pH 8.0, 150 mM NaCl, 1 mM EDTA, 1% Triton X-100, 0.1% SDS, protease inhibitor cocktail) and sonicated to shear DNA to an average size of 200–700 bp. Samples were sonicated on JY92-IIDN Ultrasonic Homogenizer (Scientz) with the following conditions: total output 10%, 1-s on, 2-s off, for 39 min on ice. Insoluble cellular debris was removed by centrifugation (14,000 rpm, 25 min, and 4 °C) and the supernatant was transferred to clear tubes. For preparing the ChIP-DNA, the protein A beads (Smart-Lifesciences, catalog^#^: SA015C) were blocked at 4 °C in IP buffer for four hours with 10 μL anti-Flag monoclonal antibody (Cat#: AGM12165, Aogma), and approximately 1.0 mL of the supernatant was incubated overnight at 4 °C on a rotating wheel with protein A beads and 10 μL anti-Flag antibodies. The protein A beads were then collected and washed once with immunoprecipitation buffer, once with immunoprecipitation buffer plus 500 mM NaCl, once with immunoprecipitation buffer plus 500 mM LiCl, and once with Tris-EDTA buffer (pH 8.0). Then, 100 μL elution buffer 1 (50 mM Tris-HCl pH 8.0, 10 mM EDTA, 1% SDS) was added to the beads and incubated at 65 °C for 20 min. The supernatants were saved after spin. After which, 150 μL elution buffer 2 (0.67% Tris-EDTA buffer) was added into the remaining beads. The elution buffer 1 and elution buffer 2 were combined, and then incubated for overnight at 65 °C to reverse crosslinking. The supernatants were then treated with RNase A by incubation for 1 h at 37 °C and 2 h at 65 °C in elution buffer plus 100 µg proteinase K. DNA was extracted twice with phenolchloroform, precipitated and re-suspended in 20 μL of water, named ChIP-DNA. For preparing the Input-DNA, 200 μL of supernatant was incubated with protein A beads in the absence of antibody, and then a similar strategy described as above was used.

For BfmR ChIP under LPM-24 h condition, we performed the experiments with two biological replicates. Overnight LPM cultures of Δ*bfmRS*::*bfmR*-*flag* strain were diluted to OD_600_ ≈ 0.05 with fresh LPM medium and grown in a 250 mL Erlenmeyer flask with a flask volume-to-medium volume ratio of 5:1, shaking with 250 rpm at 37 °C. After culturing for 24 h, a similar strategy as described above was used to perform the ChIP experiments.

For BfmR ChIP under PB-6h condition, we performed the experiments with two biological replicates. Overnight PB cultures of Δ*bfmRS*::*bfmR*-*flag* were diluted to OD_600_ ≈ 0.05 with fresh PB medium and grown in a 250 mL Erlenmeyer flask with a flask volume-to-medium volume ratio of 5:1, shaking with 250 rpm at 37 °C. After culturing for 6 h, the ChIP experiment was performed with the similar strategy as describe above.

To perform CzcR ChIP, a Δ*czcRS*::*czcR*-*flag*/p-*czcS* strain (Δ*czcRS* mutant harboring p-*czcS* plasmid and integrated vector mini-*czcR*-*flag*) was constructed. For generating mini-*czcR*-*flag* (Appendix A), a ~1.2 kb PCR product covering the region from 512 bp upstream and the *czcR* gene (not including the stop codon) was amplified using the primer pair mini-*czcR-flag*-F/R (*Hin*dIII/*Bam*HI) (Appendix A), and then the PCR products were cloned into the integrated vector mini-CTX-*lacZ*. For generating the Δ*czcRS*::*czcR*-*flag*/p-*czcS* strain, plasmid p-*czcS* was electroporated into the Δ*czcRS* mutant, yielding Δ*czcRS*/p-*czcS* strain, and then the mini-*czcR*-*flag* was further integrated into the *attB* site though a diparental mating using *E. coli* S17 λ-pir as the donor [11] (Appendix A). After that, overnight LB cultures of Δ*czcRS*::*czcR*-*flag*/p-*czcS* strain were diluted to OD_600_ ≈ 0.05 with fresh LB medium and the diluted cultures were grown in a 250 mL Erlenmeyer flask with a flask volume-to-medium volume ratio of 5:1, shaking with 250 rpm at 37 °C for 3 h. A similar strategy as described above was used to perform CzcR ChIP with three biological replicates.

### 2.6. ChIP-seq Library Construction, Sequencing, and Data Analysis

DNA fragments (150–300 bp) were selected for library construction, and sequencing libraries were prepared using the NEBNext Ultra II DNA Library Prep kit (NEB, catalog^#^: E7103). The final DNA libraries were validated with Agilent 2100 Bioanalyzer (Agilent Technologies, Palo Alto, CA, USA), then the libraries were sequenced using the HiSeq X ten system (Illumina). Each ChIP-seq experiment generated about 20 million reads, and subsequently, the ChIP-seq reads were mapped to the *P. aeruginosa* PAO1 genomes using Bowtie (Version 2.26) [32] with default parameter. The enriched peaks were identified using Model-based Analysis of ChIP-seq 2 (MACS2) software [33], which was followed by Multiple EM for Motif Elicitation (MEME) analyses [33] to generate the BfmR-binding or the CzcR-binding motif. A tiled data file (TDF) file was also created for each sample for visualization in Integrative Genomics Viewer (IGV) (Broad Institute, version 2.4.6). We filtered peaks called by MACS2 by requiring an adjusted score (i.e., -log10 *p*-value) of at least 50 in order to ensure that we had a high quality peak annotation, and peaks with a fold enrichment lower than 3-fold changes were also filtered out. The BfmR ChIP-seq data files have been deposited in National Center of Biotechnology Information’s Gene Expression Omnibus (GEO) and can be accessed through GEO Series accession number GSE154264, with the following BioSample accession numbers: GSM4668034 to GSM4668042. The CzcR ChIP-seq data files can be accessed through GEO Series accession number GSE154265, with the following BioSample accession numbers: GSM4668046 to GSM4668049.

### 2.7. Motif Detection

To study the conserved sites of BfmR binding DNA, sequences of 101 bp centered on the peak summit covering the top 30 or all 174 peaks were used for motif analysis using MEME [34] with parameters as default. For studying the CzcR binding sites, sequences of 101 bp centered on the peak summit covering the top ten peaks were used for motif analysis using MEME with default parameters except that the minimum width was set as 3.

### 2.8. RNA-seq and Data Analysis

Overnight LPM cultures of *P. aeruginosa* strains were washed three times (adjusted to OD600 ≈ 1.0 with fresh LPM) and diluted 1:50 into 20 mL LPM in a 100 mL flask. The liquid cultures were grown at 37 °C with shaking (250 rpm). After 24 h incubation, the cells were collected. Total RNA was immediately stabilized with RNA protect Bacteria Reagent (Qiagen) and then extracted by using a Qiagen RNeasy kit (catalog^#^: 74104) following the manufacturer’s instructions. Ribosomal RNA removal, cDNA library construction, and paired-end sequencing with the Illumina HiSeq2500 were completed by Biomarker Technologies CO., LTD (Beijing, China). The DESeq Package was used to detect differentially expressed genes (DEGs) [35]. A fold change ≥2 and a false discovery rate (FDR) ≤ 0.01 were used as threshold to determine the DEGs. All RNA-seq data (two biological replicates for Δ*bfmS* and Δ*bfmRS*, respectively) have been submitted to the NCBI Sequence Read Archive (SRA) (https://ncbi.nlm.nih.gov/sra/) (accessed on 10 December 2019) under the BioProject accession PRJNA504928, with the BioSample accession numbers SAMN10408230 and SAMN10408231.

### 2.9. ChIP-Quantitative Polymerase Chain Reaction (ChIP-qPCR)

ChIP-qPCR was carried out in the Bio-Rad 96 well Real-Time PCR System and 1 μL of each ChIP-DNA and Input-DNA dilution from ChIP assay with the Hieff^®^ qPCR SYBR Green Master Mix (YEASEN, Lot:H7901050) and 200 nM primers following the manufacturer’s instructions. Dissociation curve analysis was performed for verification of product homogeneity. The gene-specific primer pairs used for ChIP-qPCR for *czcR*, *phzA1* and *oprD* are RT-*czcR*-F/RT-*czcR*-R, RT-*phzA1*-F/RT-*phzA1*-R, RT-*oprD*-F/RT-*oprD*-R, respectively (Appendix A). The coding sequence of *phoB* (RT-*phoB*-F/RT-*phoB*-R) was used as a negative control. ChIP-DNA enrichment levels of interest genes were calculated by the relative quantification method (2^−ΔΔCt^ method) [36,37,38,39] and reported as fold-change. The statistics comparison was done by Student’s two-tailed *t*-test in GraphPad Prism (version 7.0).

### 2.10. Electrophoretic Mobility Shift Assay (EMSA)

The electrophoretic mobility shift experiments were performed as described in our previous studies with some modifications [11,40]. Briefly, 12 μL of the DNA probe mixture (30 to 50 ng) and purified proteins in binding buffer (10 mM Tris-Cl, pH 8.0; 1 mM DTT; 10% glycerol; 5 mM MgCl_2_; 10 mM KCl) were incubated for 30 min at 37 °C. 50 mM acetyl phosphate was added to the solution. Native polyacrylamide gel (6%) was run in 0.5 × TBE buffer at 85 V at 4 °C. The gel was stained with GelRed nucleic acid staining solution (Biotium) for 10 min, and then the DNA bands were visualized by gel exposure to 260-nm UV light.

DNA probes were PCR-amplified from *P. aeruginosa* MPAO1 genomic DNA using the primers listed in Appendix A, and the DNA fragments were listed in Appendix A.

All PCR products were purified by using a QIAquick gel purification kit (QIAGEN, catalog^#^: 28104).

### 2.11. Dye Primer-Based DNase I Footprinting Assay

The DNase I footprinting assays to determine BfmR-binding sites were performed as previously described with some modifications [11]. Briefly, PCR was used to generate FAM (carboxyfluorescein)-labeled DNA fragments. Subsequently, a 50-μL reaction mixture containing 300 ng FAM-labeled DNA and 6 μM His_6­-_BfmR (or indicated) and binding buffer (10 mM Tris-Cl, pH 8.0; 1 mM DTT; 10% glycerol; 5 mM MgCl_2_; 10 mM KCl; 50 mM acetyl phosphate) up to a volume of 50 μL was incubated at room temperature for 30 min. 0.01 unit of DNase I (Promega Biotech Co., Ltd, Cat#:137017) was added to the reaction mixture and incubated for an additional 5 min. The DNAse I digestion was terminated by adding 90 μL of quenching solution (200 mM NaCl, 30 mM EDTA, 1% SDS), and then the mixture was extracted with 200 μL of phenol-chloroform-isoamyl alcohol (25:24:1). The digested DNA fragments were isolated by ethanol precipitation. Then, 5.0 μL of digested DNA was mixed with 4.9 mL of HiDi formamide and 0.1 mL of GeneScan-500 LIZ size standards (Applied Biosystems). A 3730XL DNA analyzer detected the sample, and the result was analyzed with GeneMapper software. The dye primer based Thermo SequenaseTM Dye Primer Manual Cycle Sequencing Kit (Thermo, Lot:4313199) was used in order to more precisely determine the sequences of the BfmR protection region after the capillary electrophoresis results of the reactions were aligned, and the corresponding label-free promoter DNA fragment was used as template for DNA sequencing. Electropherograms were then analyzed with GeneMarker v1.8 (Applied Biosystems). 

For the DNase I footprinting assay of the *czcC*-*czcR* intergenic region, a 372-bp FAM-labeled promoter DNA (nucleotide −434 to nucleotide −63 relative to the start codon of *czcR*) was generated using the primer pair *czcR*-FT (BfmR)-F (FAM)/*czcR*-FT (BfmR)-R. For the DNase I footprinting assay of *phoB* promoter DNA, a 339-bp FAM labeled promoter DNA (nucleotide −199 to nucleotide −140 relative to the start codon of *phoB*) was generated using the primer pair *phoB*-FT-F/*phoB*-FT-R(FAM). For the DNase I footprinting assay of *oprP* promoter DNA, a 287 bp FAM-labeled promoter DNA (nucleotide −318 to nucleotide −32 relative to the start codon of *oprP*) was generated using the primer pair *oprP*-FT-F/*oprP*-FT-R(FAM). For the DNase I footprinting assay of *oprO* promoter DNA, a 421 bp FAM-labeled promoter DNA (nucleotide −421 to nucleotide -1 relative to the start codon of *oprO*) was generated using the primer pair *oprO*-FT-F/*oprO*-FT-R(FAM).

A similar strategy as described above was used to perform the DNase I footprinting assays for the identification of the CzcR-binding sites. Briefly, 300 ng promoter DNA and CzcR-His_6_ (18 μM or indicated) were used in reactions. For the DNase I footprinting assay of the *czcC*-*czcR* intergenic region, a 401-bp FAM-labeled promoter DNA (nucleotide −392 to nucleotide +9 relative to the start codon of *czcR*), was generated using the primer pair *czcR*-FT (CzcR)-F (FAM)/*czcR*-FT (CzcR)-R. For the DNase I footprinting assay of *phzA1* promoter DNA, a 410-bp FAM labeled promoter DNA (nucleotide −491 to nucleotide −82 relative to the start codon of *phzA1*), was generated using the primer pair *phzA1*-FT-F (FAM)/*phzA1*-FT-R.

### 2.12. Monitoring Gene Expression by lux-Based Reporters

The plasmids *czcC-lux*, *czcR-lux,* and *exsA*-*lux* (Appendix A) were conjugated into *P. aeruginosa* strains and the construct was integrated into the *attB* site as described previously though a diparental mating using *E. coli* S17 λ-pir as the donor [11]. Parts of the mini-CTX-*lux* vector containing the tetracycline resistance cassette in *P. aeruginosa* were deleted using a flippase (FLP) recombinase encoded on the pFLP2 plasmid [41].

The expression of promoter fusion genes was also carried out using a flask culture method. Briefly, overnight M8-glutamate minimal medium cultures of *P. aeruginosa* strains were diluted to OD_600_ of 0.05 with fresh minimal medium, and the diluted cultures were further incubated in a 100 mL conical flask with a volume-to-medium ratio of 5:1, shaking with 250 rpm at 37 °C. Promoter activities at 6 h of bacterial growth were measured as counts per second (CPS) of light production with a Synergy 2 Multi-Mode Microplate Reader as described previously [7,11]. Relative light units were calculated by normalizing CPS to OD_600_. The statistics comparison was done by Student’s two-tailed *t*-test in GraphPad Prism (version 7.0).

### 2.13. Western Blot Analysis

The Western blot analysis were performed as described previously with some modifications [42,43,44]. Generally, to examine the production of fusion proteins (i.e., *p*-PhoB-Flag, *p*-(D)-PhoB-Flag) in *P. aeruginosa*, overnight LPM cultures of the indicated strains were diluted to OD_600_ ≈ 0.05 with fresh LPM medium. The diluted cultures were grown in a 20 mL tube with a tube volume-to-medium volume ratio of 5:1, shaking with 250 rpm at 37 °C for about 6 h. Then approximately 1.5 × 10^9^ cells were harvested, pelleted by centrifugation at 14,000 rpm for 5 min. The samples were resuspended in 100 μL solution containing distilled water and mixed with 5 x sodium dodecyl sulfate (SDS)-polyacrylamide gel electrophoresis (PAGE) loading buffer (250 mM Tris-HCl, pH 6.8; 2% SDS; 0.1% bromophenol blue; 4% mercaptoethanol; 50% glycerol; 100 mM dithiothreitol (DTT)) and then heated at 100 °C for 15 min. Then, 10 μL sample as described above was loaded onto a 10% polyacrylamide gel, and then, SDS-PAGE was carried out at 90 V for 10 min followed by 120 V for 90 min. The proteins on gel were transferred to polyvinylidene fluoride (PVDF) (Bio-Rad, Cat^#^: 1620171) membranes through Semi-Dry Electrophoretic Transfer Cell (Bio-Rad) for 30 min at room temperature. Then, the membrane was incubated with the primary antibody followed by the secondary antibody. The antibodies were diluted with TBST buffer (10 mM Tris/HCl, pH 7.5, 150 mM NaCl, and 0.1% Tween 20) containing 5% (wt/vol) skimmed milk powder.

PhoB-Flag proteins were detected by Western blot analysis using a mouse anti-Flag monoclonal antibody (Aogma, catalog^#^: AGM12165) followed by a secondary, sheep anti-mouse IgG antibody conjugated to horseradish peroxidase (HRP) (GE Healthcare, Code^#^: NA931). For detection of RNAP protein, anti-RNAP (Neoclone, #WP003) antibody and anti-mouse IgG antibody conjugated to horseradish peroxidase (HRP) (GE Healthcare, Code^#^: NA931). Immunoblots for RNAP served as loading control. The images were taken using Tanon-5200 Multi (Tanon, Shanghai, China), according to the manufacturer’s recommendation. When appropriate, the relative abundance was determined by densitometric analysis using the ImageQuant software (Molecular Dynamics, Sunnyvale, CA, USA) and the expression was normalized to the indicated loading control (results are reported as fold changes with control bacteria set to 1, as indicated).

### 2.14. Measurement of Inorganic Phosphate Level and Bacteria Growth in the Medium

Firstly, we prepared a standard phosphate solution contain 2 μg of potassium dihydrogen phosphate (KH_2_PO_4_) per 1 mL. We added 0, 5, 10, 30, 50, 100, and 150 μL of standard phosphate solution to Eppendorf (EP) tubes, then respectively diluted into 500 μL with distilled water. Subsequently, we mixed 500 μL of dilution with 10 μL of 10% ascorbic acid and incubated for 30 s at room temperature, 20 μL ammonium heptamolybdate was added and incubated at room temperature for about 15 min. Then we measured the absorbance at 700 nm and drew a KH_2_PO_4_ standard curve using zero phosphate solution as the control.

A similar strategy as described above was used to measure the inorganic phosphate level and bacteria growth in the medium. Overnight LPM cultures of *P. aeruginosa* strains were washed three times with fresh LPM medium and diluted into 20 mL LPM in a 100 mL flask (adjusted to OD_600_ ≈ 0.05). The liquid cultures were grown at 37 °C with shaking (250 rpm). After 24 h of culturing, a 100 μL sample was added to the 96-well plate with transparent bottom. The absorption of OD_600_ was detected by a Synergy 2 Multi-Mode Microplate Reader. Meanwhile, 1 mL of the culture was centrifuged, and the supernatant was transferred to EP tubes. Then we mixed 500 μL supernatant with 10 μL of 10 % ascorbic acid at room temperature for 30 s, 20 μL ammonium heptamolybdate was added and incubated at room temperature for about 15 min. Then, 100 μL sample was transferred to a 96-well plate, and the absorbance at 700 nm was measured with the plate reader. The inorganic phosphate concentration of each sample was calculated by comparison to the KH_2_PO_4_ standard curve. The statistics comparison was done by Student’s two-tailed *t*-test in GraphPad Prism (version 7.0).

### 2.15. Measurement of Intracellular Polyphosphate (polyP)

Overnight LPM cultures of *P. aeruginosa* strains were washed three times with fresh LPM medium and diluted into 20 mL LPM in a 100 mL flask (adjusted to OD_600_ ≈ 0.05). After incubation at 37 °C for 24 h, 10 mL cultures were harvested, and the cell pellets were washed three times with 100 mM Tris-HCl (pH 7.4). 4′,6-diamidino-2-phenylindole (DAPI) at 10 μM was added to 2 mL cell suspensions (OD_600_ ≈ 0.2) in 100 mM Tris-HCl (pH 7.4). After 5 min of agitation at room temperature in the dark, the DAPI fluorescence spectra (excitation: 450 nm; emission: 450 to 650 nm) were recorded using a Synergy 2 Multi-Mode Microplate Reader. The fluorescence of the DAPI-polyP complex at 550 nm was used to measure intracellular polyP concentration because fluorescence emission from DAPI and DAPI-DNA are minimal at this wavelength [45]. The polyP level was calculated by normalizing fluorescence to OD_600_. The statistics comparison was done by Student’s two-tailed *t*-test in GraphPad Prism (version 7.0).

### 2.16. Biofilm Formation Assays

The biofilm formation were measured by detecting the ability of the cells to attach to the wells of polystyrene Stripwell^TM^ Microplate (1 X 8 Flat Bottom; Corning incorporated, Costar, Code#: 42592) as previously described with minor modifications [7,46]. Briefly, an overnight PB culture was diluted to a final OD_600_ of 0.05 in fresh PB medium and dispensed 100 μL into per well. The plates were incubated under static conditions for 72 h at 37 °C. In order to measure the degree of attachment, non-adhered cells were removed, and the biofilms rinsed with distilled water. Biofilms were stained by 150 μL of 1% crystal violet (Cat#: 3603, Sigma-Aldrich) for 15 min. Photos were taken and crystal violet was solubilized in 150 μL of 30% acetic acids, the extent of biofilm formation was quantified by measuring the OD_595_ of the resulting solution. The statistics comparison was done by Student’s two-tailed *t*-test in GraphPad Prism (version 7.0).

## 3. Results

### 3.1. Genome-Wide Binding Patterns of BfmR in P. aeruginosa MPAO1

To explore the BfmR regulon in *P. aeruginosa*, we performed ChIP-seq analysis with a Δ*bfmRS*::*BfmR*-*Flag* strain [11] expressing a functional C-terminally tagged BfmR protein (BfmR-flag) under three following growth conditions: (1) in defined low-phosphate minimal (LPM) medium for 6 h (LPM-6 h), (2) in LPM medium for 24 h (LPM-24 h), (3) in complex pyocyanin production broth (PB) medium for 6 h (PB-6 h). Using the peak-calling software MACS2, we identified 172 reproducible enriched regions (fold enrichment ≥ 3, *p*-value ≤ 1 × 10^-5^) in the LPM-6 h samples, while 76 reproducible enriched regions were found in LPM-24h samples and only 16 were identified in the PB-6h samples (Data set 1, sheet 1–6).

The ChIP-seq peaks for BfmR revealed broad occupancy over the *P. aeruginosa* PAO1 chromosome (Figure 1). According to the ChIP-seq data under the three different bacterial growth conditions, the top three most enriched regions were located in the promoter regions of genes (i.e., *bfmR*, *pa4103*, and *pa4107*) that were previously shown to be directly regulated by BfmR (Figure 1B; Data set 1, sheet 2, 4, and 6), supporting the effectiveness of the ChIP-seq procedure. Most of the binding sites (74 out of 76) occupied under LPM-24 conditions overlap those under LPM-6 conditions (Data set 1, sheet 4). Comparison of enriched ChIP-seq peaks revealed that 16 peak regions are common between the three growth conditions (Appendix A, Data set 1, sheet 6). In total, 174 BfmR-enriched regions were identified (Appendix A, Data set 2).

To identify the BfmR-binding motif, we searched the 101-bp sequences centered around each peak summit using the Multiple EM for Motif Elicitation (MEME) web tool [47]. With the top 30 peak sequences for the LPM-6h sample (Data set 1, sheet 1), we identified a 14-bp consensus sequence characterized by a conserved motif TACAA-N_3_-GATACA (Figure 1C), while only half of this motif (5′-GATACA-3′) was detected when we used all of the 174 peak sequences (Appendix A). In each case, the MEME algorithm located the motif in the promoters of *bfmR* and *pa4103* at the position of the BfmR-binding sites previously identified by DNase I footprinting [11] (Figure 1C; Appendix A), suggesting that the determination of BfmR-binding consensus sequence was carried out successfully. Out of the 174 enriched peaks (Data set 2), 161 (93%) displayed 101 bp fragment centered on the peak summits located at intergenic regions (Appendix A, Data set 2), indicating a high enrichment of BfmR-binding sites at regulatory regions since approximately 90% of the *P. aeruginosa* MPAO1 genome is composed of coding sequences [48].

### 3.2. Identification of BfmR-Regulated Genes

Since some enriched regions are flanked by divergently transcribed genes (or presumptive operons), BfmR binding within those 174 regions could potentially control the expression of 409 genes in 239 predicted TUs (Data set 2), given that bacterial genes can be remotely regulated (over 1 kb upstream of the translation start site) by transcriptional regulators [49]. For the promoter of each potential BfmR-targeted TUs, we determined the distance between the translational start site and the corresponding BfmR ChIP-seq peak summit, obtaining a binding distribution that pointed to a preferential localization between 20 and 300 bp upstream the start codons (Data set 2, sheet 1 and 2). Interestingly, BfmR bound to the promoters of genes encoding a variety of transcriptional regulators including CzcR, ExsA, PhoB, NalC, and MvfR (PqsR) (Data set 2), suggesting that BfmR may have a profound effect on the gene regulatory networks in *P. aeruginosa*. 

Very recently, we have identified 826 BfmR-regulated genes in *P. aeruginosa* MPAO1 grown in M8-glutamate minimal medium at 37 °C for 6 h [7]. To further define the BfmR-regulated genes, we compared the transcript levels between Δ*bfmS* mutant (deficient in *bfmS*) and Δ*bfmRS* mutant (deficient in both *bfmR* and *bfmS*) grown in LPM medium at 37 °C for 24 h, a culture condition which allows BfmR to exert strong repressing activity against the Rhl QS system of *P. aeruginosa* [11]. As a result, a total of 1639 genes were differentially expressed (≥2-fold change, FDR ≤ 0.01) (Data set 3), indicating that the effect of BfmR on the expression of *P. aeruginosa* genes was much more profound than we previously thought. Indeed, integration of these results with our previously published transcriptome data sets [7,11] revealed that BfmR is capable of directly or indirectly modulating the expression of 2150 genes, representing 37.7% of the total number of annotated genes (i.e., 5697) in the *P. aeruginosa* PAO1 genome.

To characterize genes under the direct control of BfmR, we compared the 409 potential BfmR-targeted genes obtained from ChIP-seq experiments (Data set 2) with RNA-seq results. We observed that a total of 172 genes in 106 predicted transcription units appear to be directly regulated by BfmR (Figure 1D, Data set 2). Moreover, there are 79 chromosomal regions at which BfmR binding was observed but for which RNA-seq experiments revealed no genes was significantly influenced by BfmR (Data set 2). Possible explanations for this observation are (i) the binding by BfmR was not physiologically significant; (ii) the gene was differentially expressed but did not meet the criteria for selecting differentially expressed genes.

Collectively, by combining ChIP-seq data and RNA-seq analysis, we found that a total of 172 BfmR-targeted genes represents high-confidence members of the BfmR regulon and includes known target genes such as *pa4103*, *pa4104*, *pa4105*, *pa4106*, and *pa4107* (Data set 2). Functional analyses indicate that those BfmR-targeted genes are involved in different biological processes including oxidative phosphorylation (e.g., *cyoABCDE* operon), metabolism (e.g., *phhABC* operons), antibiotic resistance (e.g., *mexB*, *oprM*, *nalC*, and *mexJK* operon), virulence (e.g., *lasA*, *aprA*, *phzC1*, and *phzG1*), phosphate uptake (e.g., *phoB*), and quorum sensing (e.g., *pqsABCDE* operon and *mvfR*) (Data set 2).

### 3.3. BfmR Binds to the Promoters of CzcR-Targeted Genes

Among the 174 BfmR-binding sites identified by ChIP-seq experiments, 4 are located in the promoters of well-known CzcR targets (i.e., *czcR*, *czcCBA* operon*, oprD*, and *phzA1*) (Data set 1, sheet 2 and 8). In *Pseudomonas* species, CzcR is the response regulator of zinc-responsive CzcRS TCS that confers resistance to zinc, cadmium, and cobalt, and antibiotic imipenem [50,51,52,53,54]. Intrigued by these observations, we sought to investigate to what extent the targets overlap between BfmR and CzcR. To accomplish this, we performed the ChIP-seq experiments to determine the CzcR-binding sites. We found that 16 chromosomal regions were enriched (fold enrichment ≥ 3, *p*-value ≤ 1 × 10^-5^) by coimmunoprecipitation with a C-terminally tagged CzcR protein from all three biological replicates under growth condition of 37 °C for 3 h in Luria-Bertani (LB) broth supplemented with ZnCl_2_ (Figure 2A; Data set 1, sheet 7 and 8). Chromatin immunoprecipitation-quantitative polymerase chain reaction (ChIP-qPCR) experiments showed that CzcR binds to the promoter of *czcR*, *phzA1*, and *oprD* (Appendix A), and this is consistent with the ChIP-seq results (Figure 2A). Moreover, using electrophoretic mobility shift assay (EMSA) and DNase I Footprinting assay, we validated that both *phzA1* promoter and *czcC*-*czcR* intergenic region are bound by CzcR (Appendix A).

Using MEME, we identified a 16-bp motif (i.e., GAAAC-N_6_-GTAAT) (Figure 2B) that was found in each of the top10 enriched peaks (average fold enrichment ≥ 10) (Data set 1, sheet 7). Inspection of the alignment reveals similarity to the CzcR conserved motif ATTAC-N_6_-GTAAT reported previously in *Pseudomonas stutzeri* [53]. In this case, the AC-N_6_-GTAAT was highly conserved, indicating that it may represent the core binding site for CzcR in these two different *Pseudomonas* species.

Annotation of the 16 CzcR-enriched regions according to their positions showed that a total of 47 genes in 24 predicted TUs is likely to be directly regulated by CzcR (Data set 4, sheet 1 and 2). Interestingly, 11 out of the 24 CzcR-targeted promoters were also bound by BfmR (Data set 4, sheet 1). Moreover, the ChIP-seq signal of BfmR was also observed for CzcR-targets such as the promoters of *oprO* (with an average fold-enrichment of 2.2) (Appendix A), *pa4138* (with an average fold-enrichment of 2.0) (Appendix A), *pa4142* (with an average fold-enrichment of 2.1) (Appendix A), and *pa2936* (with an average fold-enrichment of 2.7) (Appendix A), although it does not meet the threshold (fold enrichment ≥ 3 and *p*-value ≤ 1 × 10^-5^) for defining the BfmR-targeted sites. Nonetheless, these results suggest that BfmR regulon overlaps with the CzcR regulon extensively.

Using EMSA and DNase I Footprinting assays, we verified that BfmR is able to bind to the intergenic region of *czcR* and *czcC* (Figure 3A–D). We also observed that when compared to wild-type MPAO1, the Δ*bfmS* mutant but not the Δ*bfmRS* mutant exhibited a decreased promoter activity of both *czcC* and *czcR* (Figure 3E,F), which implies that BfmR might directly negatively control the expression of *czcCBA* and *czcRS*. Moreover, our RNA-seq experiments showed that additional 9 CzcR-target genes (i.e., *pa0959*, *czcR*, *pa2696*, *oprO*, *algE*, *pa4139*, *phzM*, *phzC1*, and *phzG1*) are regulated by BfmR (Data set 4, sheet 3), indicating that BfmR may have a profound effect on the expression of the CzcR regulon in a CzcR-dependent and -independent manner.

### 3.4. BfmR Binds to and Induces exsA Promoter

As aforementioned, BfmR binds to the promoters of a number of genes encoding regulatory proteins including ExsA (Data set 1, sheet 2), which is the central regulator of T3SS gene expression and is encoded by the last gene in the *exsCEBA* operon [55,56]. The expression of *exsA* is driven by a promoter upstream of *exsC*; however, the intergenic region between *exsB* and *exsA* also displayed promoter activity [55,56,57]. In our ChIP-seq experiments, we showed that BfmR directly bound to the region immediately upstream of *exsA* with an average fold-enrichment larger than 10 in LPM-6h sample (Appendix A, Data set 1, sheet 2). In line with this observation, our EMSA assays also showed that BfmR binds to a DNA sequence locating 366 bp upstream of the *exsA* (Appendix A).

To determine whether the expression of *exsA* is regulated by BfmR in *P. aeruginosa*, we constructed an *exsA* promoter-*lux* fusion (*exsA*-*lux*, Appendix A) and measured its activity in *P. aeruginosa* wild-type MPAO1, the Δ*bfmS* mutant, and the Δ*bfmRS* mutant. The activity of *exsA-lux* in the Δ*bfmS* mutant was about fourfold higher than that of either the wild-type MPAO1 or the Δ*bfmRS* mutant (Appendix A). Introduction of p-*bfmS* plasmid into the Δ*bfmS* mutant restored the *exsA-lux* activity to the wild-type level (Appendix A). Additionally, we found that complementation of Δ*bfmRS* double mutant with a plasmid-borne *bfmR* (i.e., p-*bfmR*) causes a twofold increase in the activity of *exsA-lux* (Appendix A). Based on these results, we concluded that BfmR can activate the expression of *exsA* directly, although our RNA-seq experiments failed to identify *exsA* as a BfmR-regulated gene (Data set 2) [7]. These observations also suggest that the number of BfmR-targeted genes revealed by ChIP-seq and RNA-seq experiments (Data set 2) might be underestimated.

### 3.5. BfmR Binds to the Promoter of a Number of PhoB-Targeted Genes

Our ChIP-seq experiments showed that BfmR binds to several well-known PhoB-targeted genes including *phoB*, *oprO*, *oprP*, *pdtA*, *mvfR*, and *phzA1* (Figure 4; Data set 4, sheet 1) [59,60,61,62,63]. In *P. aeruginosa*, PhoB is the response regulator of PhoBR TCS that modulates genes expression in response to Pi-limiting environments [64,65]. Currently, a total of 180 genes in 164 TUs have been characterized as PhoB regulon in *P. aeruginosa* PAO1 based on strong experimental and computational evidence [58,59,64] (Data set 4, sheet 1 and 4). To further investigate the relationship between the BfmR and PhoB regulons, we examined the BfmR ChIP-seq signals in the promoter of PhoB-targeted transcription (i.e., gene or operon). We observed that the promoter regions of 28.6% (47 out of 164) of PhoB-targeted transcription units were bound by BfmR (Figure 4, Data set 4, sheet 1). Furthermore, ChIP-seq signal of BfmR was also observed for the promoters of at least 34 PhoB-targeted transcription units (e.g., *amrZ*, *vreAIR* operon, and *oprO*) (Data set 4, sheet 1); although, it does not meet the threshold for defining BfmR targets. These findings suggest that there is a wide distribution of DNA binding sites for BfmR in the promoters of PhoB-targeted genes in *P. aeruginosa*.

In line with the results obtained with ChIP-seq experiments (Figure 5A; Appendix A), EMSA and DNase I Footprinting assays showed that BfmR does bind to the promoters of the well-known PhoB-targeted genes including *phoB*, *oprO*, and *oprP* (Figure 5B, C; Appendix A). In addition, we observed that 66 out of the 180 PhoB regulon genes exhibited differentially expressed level in Δ*bfmS* mutant compared to the Δ*bfmRS* mutant (Data set 4, sheet 4 and 5), suggesting that BfmR has profound effect on the expression of PhoB-targeted genes. 

### 3.6. BfmR Directly Activates phoB

To examine whether BfmR influences the activity of *phoB* promoter, we performed Western blot analysis of the expression of Flag-tagged PhoB, driven by its native promoter, in wild-type MPAO1 strain, Δ*bfmS* mutant, its complemented strain (Δ*bfmS*/p-*bfmS*), and the Δ*bfmRS* mutant. The Δ*bfmS* mutant strain exhibited about ninefold higher PhoB production than the wild-type strain (Figure 5D). Complementation of the Δ*bfmS* mutant with a plasmid-borne *bfmS* (Δ*bfmS*/p-*bfmS*) decreased the production of PhoB-Flag to wild-type levels (Figure 5D). Moreover, the production of PhoB-Flag in the Δ*bfmRS* mutant was similar to that of the wild-type MPAO1 strain (Figure 5D), suggesting that in *P. aeruginosa* the effect of *bfmS* deletion on the promoter activity of *phoB* requires BfmR. 

To determine if the BfmR-binding motif (TACAA-N_3_-GATACA) is involved in the BfmR-mediated activation of the *phoB* promoter, we deleted the six residues (i.e., GACACA) located at position 9–14 of the BfmR-binding motif and examined the ability of the mutant *phoB* promoter to permit the induction of the reporter protein PhoB-Flag in a Δ*bfmS* strain. As shown, the production of PhoB-Flag in the Δ*bfmS* strain was similar to that observed in the wild-type MPAO1 strain (Figure 5E), indicating that the six residues (GACACA) in the BfmR-binding site is crucial for BfmR-mediated regulation of *phoB.*


### 3.7. BfmR Boosts Pi Depletion and Inorganic Polyphosphate (polyP) Accumulation

To further examine the role of BfmR in the activation of *phoB*, we performed inorganic phosphate (Pi) depletion assays because the uptake of Pi is controlled by PhoB, which induces the transcription of phosphate uptake genes under low-phosphate conditions [66,67]. As can be seen in Figure 5F,G, although Δ*bfmS* mutant suffered a decrease in biomass production, it showed a 28-fold increase of Pi depletion when compared to the wild-type MPAO1 (Figure 5G). The spent medium from Δ*bfmRS* mutant culture exhibited the residual Pi levels similar to that of the wild-type MPAO1 strain (Figure 5G), indicating that BfmR increases the uptake of Pi in the Δ*bfmS* mutant. We also observed that introduction of a plasmid-borne *bfmS* into the Δ*bfmS* mutant significantly decreases the ability of Δ*bfmS* mutant to remove Pi from the medium (Figure 5G), supporting the notion that BfmS negatively regulates BfmR [7,11]. Additionally, Δ*bfmS* mutant accumulated higher levels of inorganic polyphosphate (polyP) compared to either the WT MPAO1 strain, the complemented strain of the Δ*bfmS* mutant, or the Δ*bfmS* mutant (Figure 5H), and these results are in consistent with the observation that BfmR directly upregulates *phoB* (Figure 5D,E) and that the polyphosphate is synthesized in a PhoB-dependent manner [68]. Thus, BfmR boosts Pi removal and polyP accumulation, supporting the notion that the expression of *phoB* can be activated by BfmR (Figure 5D; Data set 2).

### 3.8. PhoB Is Required for BfmR-Dependent Biofilm Formation

It has been shown that BfmR regulates biofilm development through activating *phdA* [18], a PhoB-dependent gene [61,62]. We thus asked whether PhoB has a role in regulating the biofilm formation in the Δ*bfmS* mutant, for which an enhanced biofilm formation phenotype was observed due to the activation of BfmR [11]. To this end, we created a *phoB* deletion mutant in the Δ*bfmS* background and performed biofilm formation assays in the microtiter plates with *P. aeruginosa* strains including WT MPAO1 strain, Δ*bfmS* mutant, and the Δ*phoB*Δ*bfmS* double mutant. As expected, deletion of *bfmS* alone resulted in a 2.3-fold increase in the biofilm formation, while deletion of both *bfmS* and *bfmR* has no obvious effect (Figure 6A,B). Importantly, Δ*phoB*Δ*bfmS* double mutant produced significantly lesser biofilm than the Δ*bfmS* mutant (a 2.2-fold decrease) (Figure 6A,B). Complementation of Δ*phoB*Δ*bfmS* double mutant with a plasmid-borne WT *phoB* restored biofilm formation to a level similar to that of the Δ*bfmS* (Figure 6A,B). Collectively, these results suggest that *phoB* plays an essential role in BfmR-dependent biofilm formation in *P. aeruginosa*, at least under our testing condition.

## 4. Discussions

Here, we provide the first genome-wide analysis of the regulon of *P. aeruginosa* BfmR and a broader view of the cellular processes influenced by this important RR. Our work confirms several BfmR target genes as defined previously [11,18] and significantly expands our understanding of the BfmR-regulated genes, highlighting the complexity of gene regulation by the BfmRS TCS in *P. aeruginosa* (Figure 6C). 

Using ChIP-seq experiments, we identified a total of 174 BfmR binding sites in *P. aeruginosa* MPAO1 (Data set 2). The number of BfmR binding sites are variable in different bacterial growth conditions (Data set 1, sheet 2, 4, and 6). A possible explanation for these observations is that the phosphorylation levels of BfmR may be different under these test conditions. Indeed, only 16 of 174 sites occupied by BfmR under our three test conditions (Appendix A), presumably represent sites bound by BfmR that are of particularly high affinity. Of note, the top 3 BfmR enriched regions correspond to the promoters of known BfmR-targeted transcripts including *pa4107* (*efhP*)*-pa4106-pa4105*, *pa4103-pa4104*, and *pa4101-pa4102* (*bfmRS*) operons [11], supporting that these operons are the major targets of BfmR. A weak ChIP-seq signal was also observed for the promoter of another two known BfmR targets (i.e., *rhlR* and *pdhA*), although the degree of enrichment did not meet the criteria for the selection of a BfmR binding sites (Appendix A). Therefore, the number of BfmR binding sites (Data set 2) may be underestimated in this study due to relatively stringent criteria used.

Our RNA-seq data showed that *efhP*, the first gene of the *pa4107* (*efhP*)*-pa4106-pa4105*, is the most highly activated gene by BfmR (11, 268-fold increase) (Data set 3). This result is in consistent with our previous findings [7,11]. In *P. aeruginosa*, *efhP* encodes an EF-Hand protein that is important for Ca^2+^ homeostasis and bacterial virulence [69]. It has also been reported that, like the activation of BfmR [11], calcium was capable of inducing a switch from the acute to the chronic virulence state of the *P. aeruginosa* [70]. These findings suggest that BfmR-mediated activation of *efhP* may contribute to Ca^2+^ homeostasis and thus the host adaptation by *P. aeruginosa*. Moreover, BfmR activates the promoter of *czcR* encoding the response regulator of CzcRS TCS involved in heavy metal resistance (Figure 3E) [13,50], reinforcing the likelihood that BfmRS has a profound effect on the metal homeostasis in *P. aeruginosa* (Figure 6C). 

BfmR is a negative regulator of the Rhl QS system [7,11]. In line with this, our RNA-seq data showed that a number of Rhl-regulated genes (e.g., *rhlR*, *rhlI*, and *rhlA*) were repressed by BfmR (Data set 3). Interestingly, activation of BfmR also caused an impairment of the Pqs QS system in *P. aeruginosa*, as evidenced by the fact that the expression level of *pqsABCDE* operon, which is required for the synthesis of PQS autoinducer, was significantly decreased (range from 3.7-fold to 7.4-fold) in the Δ*bfmS* mutant compared to the Δ*bfmRS* (Data set 3). Thus, BfmR-mediated inhibition of the Pqs QS system may contribute to the attenuation of Rhl QS system in the Δ*bfmS* mutant because the Rhl QS system is activated by the Pqs QS [59]. Interestingly, our ChIP-seq experiments showed that BfmR binds to the promoter of *pqsABCDE* (Appendix A), which suggests that this operon is under the direct control of BfmR.

Like in the case of Rhl QS, BfmR appears to modulate *P. aeruginosa* biofilm formation through different mechanism of actions (Figure 6C). Petrova *et al.* have reported that BfmR controls *P. aeruginosa* biofilm development through direct regulation of *phdA* [18], a PhoB-dependent gene [61,62]. In this study, we showed that BfmR activates *phoB* to promote the biofilm formation by *P. aeruginosa* (Figure 5 and Figure 6A,B). Given these, BfmR may contribute to biofilm development by activating *phdA* directly or indirectly through *phoB*, or both (Figure 6C). In addition, it has been shown that the availability of phosphate enhances biofilm formation independent of the inhibiting activity of the PhoB regulon [71]. It also has been reported that PhoB modulates *P. aeruginosa* QS systems [63,66] that affect the biofilm formation by *P. aeruginosa* [72,73,74,75]. Thus, BfmR-mediated regulation of biofilm formation and QS may be very complicated, via multiple signaling pathways that are regulated by various environmental signals (Figure 6C). Further studies are required to determine how the biofilm formation and QS are regulated by BfmR in the context of infections.

In summary, in this study we provided a snapshot of the regulatory network of BfmR and demonstrated that BfmR-mediated activation of *phoB* is a mechanism by which *P. aeruginosa* promotes the formation of biofilm (Figure 6C). Our findings also emphasize that more work needs to be done to increase our understanding of the regulation of virulence evolution and biofilm formation in *P. aeruginosa* during chronic infections. 

## Figures and Tables

**Figure 1 microorganisms-09-00485-f001:**
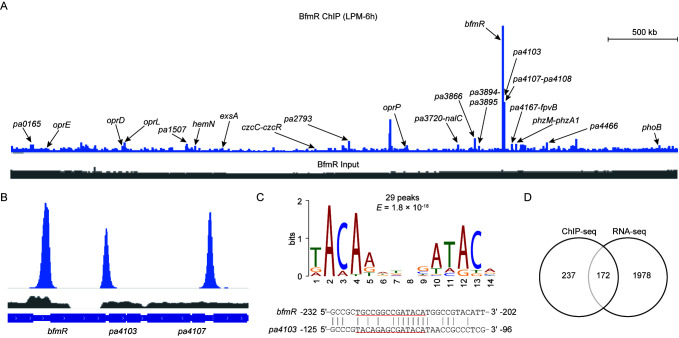
Analysis of BfmR ChIP-seq. (**A**) A representative image of BfmR ChIP-seq result using Integrated Genome Viewer. Bacteria were cultured in low-phosphate minimal (LPM) for 6 h (LPM-6h). ChIP sample (blue) and input control sample (grey) were shown as indicated. Arrows indicate the gene promoters associated with the ChIP-seq peaks. (**B**) Pattern of BfmR ChIP-seq peaks for the promoter regions of *bfmR*, *pa4103*, and *pa4107*. (**C**) The most significant motif generated by the MEME tool [34] using 101 bp centered on the peak summit of the top 30 peak sequences (Data set 1, sheet 1) with default parameter values. The height of each letter represents the frequency of each base in different locations in the consensus sequence. The 14 nt motif was present in 29 BfmR binding sites with an E-value of 1.8 × 10^-18^ (upper panel). The location of the conserved promoter element generated by MEME (*upper panel*) on the BfmR-protected region of either *bfmR* or *pa4103* promoter [11] was underlined (lower panel). (**D**) Venn diagram of integrated ChIP-seq and RNA-seq results showing the direct and indirect targets of BfmR (Data set 2).

**Figure 2 microorganisms-09-00485-f002:**
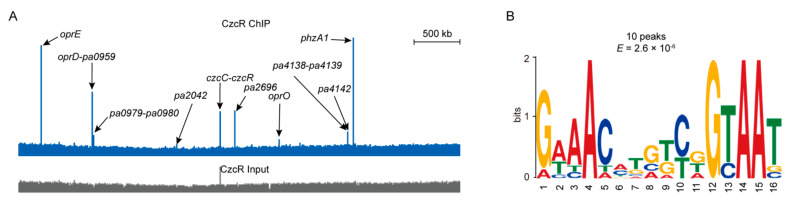
Identification of CzcR targets. (**A**) A representative image of CzcR ChIP-seq result using Integrated Genome Viewer. ChIP sample (blue) and the input control sample (grey) were shown. Arrows showing the ChIP-seq peaks located in the promoters of the indicated genes. (**B**) The most significant motif was generated by the MEME tool using 101 bp centered on the peak summits of the top 10 peak sequences (Data set 1, sheet 8), with minimum width as 3 and other parameters as default. The height of each letter represents the frequency of each base in different positions in the consensus sequence. The 16 nt motif was present in all the 10 tested CzcR binding sites with an E-value of 2.6 × 10^-6^.

**Figure 3 microorganisms-09-00485-f003:**
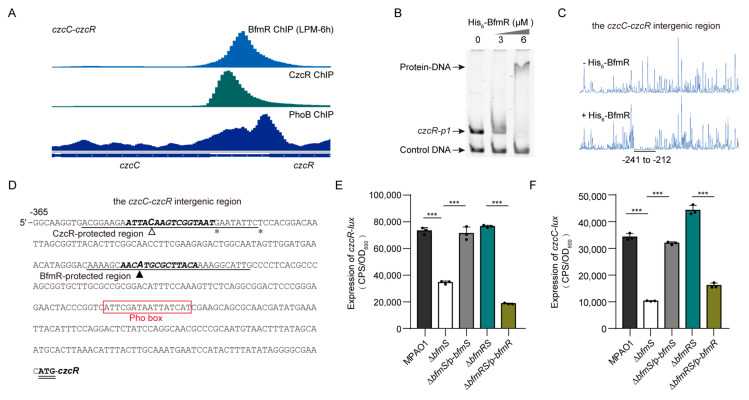
BfmR binds to and inhibits the promoter activity of *czcR* and *czcC*. (**A**) Pattern of BfmR, CzcR, and PhoB ChIP-seq peaks for the *czcC*-*czcR* intergenic region. The PhoB ChIP data were obtained from NCBI Gene Expression Omnibus (GEO) database under accession number GSE128430 [58]. (**B**) EMSA showing that the N-terminal His_6_-tagged BfmR (His_6_-BfmR) binds to the promoter of *czcR* (i.e., *czcR-p1*). DNA fragments C-*bfmR* serves as a negative control. (**C**) Electropherograms show the protection pattern of the *czcC*-*czcR* intergenic region after digestion with DNase I following incubation in the absence (-) and presence (+) of His_6_-BfmR (6 μM). The protected regions (relative to the start codon of *czcR*) are underlined. (**D**) *czcR* promoter sequence with a summary of the results of DNase I footprint assays and ChIP-seq experiments. The CzcR- and BfmR-protected regions are underlined as indicated and the asterisk shows the DNase I hypersensitivity site described in Appendix A. Hollow and solid triangle indicates the location of the summits of CzcR- and BfmR ChIP-seq peaks, respectively. The potential Pho box [59] is highlighted by square frame, and the starting codon (ATG) of *czcR* is in bold and double underlined. Sequences that match the MEME motif of CzcR (see in Figure 2B) and BfmR (see in Figure 1C) are in bold and italic. (**E**,**F**) The promoter activity of *czcR* (in E) and *czcC* (in F) in wild type (WT) MPAO1 and its derivatives grown in MM supplemented with 50 μM ZnCl_2_ at 37 °C for 6 h. Data points are shown in black dots, and results represent means ± SD (n = 3 biological replicates; *** *p* < 0.001, Student’s two-tailed *t*-test). MPAO1, Δ*bfmS*, and Δ*bfmRS* harboring an empty pAK1900 vector as a control; p-*bfmS* and p-*bfmR* respectively denote pAK1900-*bfmS* and pAK1900-*bfmR* (Appendix A).

**Figure 4 microorganisms-09-00485-f004:**
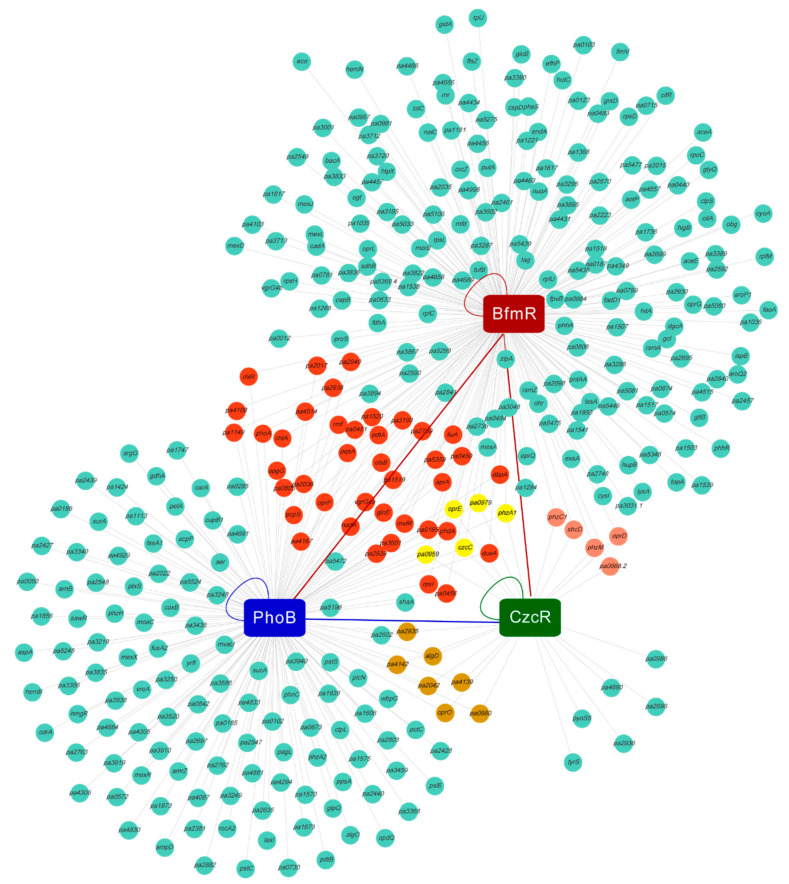
Visualization of promoters bound by BfmR, PhoB, and CzcR. Red circle denotes gene (or the first gene in an operon) whose promoter is co-regulated by BfmR and PhoB; yellow circle, gene co-regulated by BfmR, PhoB, and CzcR; pink circle, co-targeted by BfmR and CzcR; brown circle, co-targeted by PhoB and CzcR. The lines show the connection between the transcription factors and promoters. The curved line represents the binding of the transcription factor to its own promoter [11,53,58,64]. Red lines show the binding of BfmR to the promoters *phoB* and *czcR* (Figure 1B, Figure 3A and Figure 5A); Blue line show the binding of PhoB to *czcR* promoter (Figure 3A and Figure 5A). See Data set 4, Sheet 1 for details.

**Figure 5 microorganisms-09-00485-f005:**
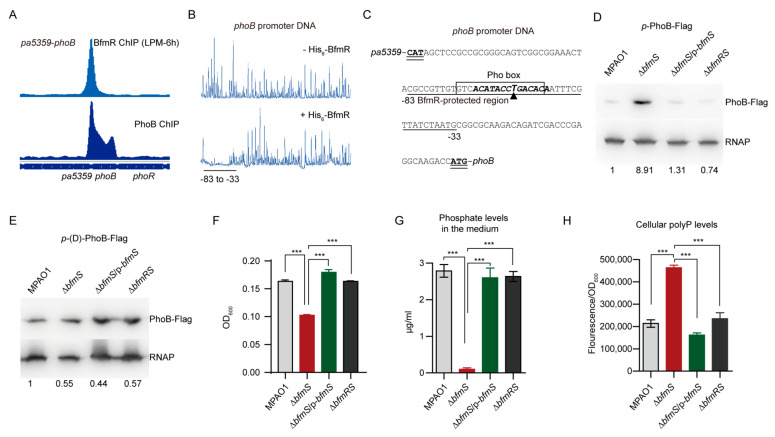
BfmR binds to and activates *phoB* promoter. (**A**) BfmR and PhoB ChIP-seq signals in *phoB* promoter. The PhoB ChIP data were obtained from GEO (accession number GSE128430) [58]. (**B**) Electropherogram shows the protection pattern of the *phoB* promoter DNA after digestion with DNase I following incubation without or with His_6_-BfmR (6 μM). The protected regions (relative to the start codon of *phoB*) are underlined. (**C**) Intergenic sequence of *pa5358* and *phoB* with a summary of the results of DNase I footprint assays and ChIP-seq experiments. The BfmR-protected region (relative to the start codon of *phoB*) is underlined; triangle showing the location of the peak summit for BfmR ChIP-seq; sequences that match the MEME motifs of BfmR (see in Figure 1C) are in bold and italic. The potential Pho box [59] is highlighted by square frame, and the starting codons of *pa5359* and *phoB* are in bold and double underlined. (**D**,**E**) Western blot assays showing the production of PhoB-Flag in *P. aeruginosa* MPAO1 and its derivatives grown in tubes containing LPM at 37 °C with shaking for 6 h. The FLAG-tagged PhoB fusion gene is under the control of a native (in D) or a mutant *phoB* promoter (in E, lack of GACACA in the BfmR-protected region). The Western blot band intensity of PhoB-Flag was normalized to the intensities obtained with RNA polymerase (RNAP) (used as a loading control) and the results are reported as fold changes with the WT MPAO1 set to 1. MPAO1, Δ*bfmS*, and Δ*bfmRS* harbor an empty pAK1900 vector as a control; p-*bfmS* denotes pAK1900-*bfmS* (Appendix A). Data are representative of three biological replicates. (**F**) The growth of *P. aeruginosa* MPAO1 and its derivatives. Bacteria were grown for 24 h in LPM medium. Results represent means ± SD (n = 3 biological replicates; *** *p* < 0.001, Student’s two-tailed *t*-test); OD_600_, an optical density at 600 nm. (**G**) Phosphate removal from the medium. Bacteria were grown for 24 h in LPM medium, and the phosphate level in the spent medium was measured. The initial level of phosphate in the LPM medium is 28 µg/mL (0.3 mM). (**H**) Measurements of the polyphosphate (polyP) level in WT MPAO1 and its derivatives grown in LPM medium at 37 °C for 24 h. The polyP was quantified using 4’,6’-diamidino-2-phenylindole (DAPI) fluorescence as described in Methods. In (F and G), MPAO1, Δ*bfmS*, and Δ*bfmRS* harbor an empty pAK1900 vector as a control; p-*bfmS* denotes pAK1900-*bfmS* (Appendix A); results represent means ± SD (n = 3 biological replicates; *** *p* < 0.001, Student’s two-tailed t-test).

**Figure 6 microorganisms-09-00485-f006:**
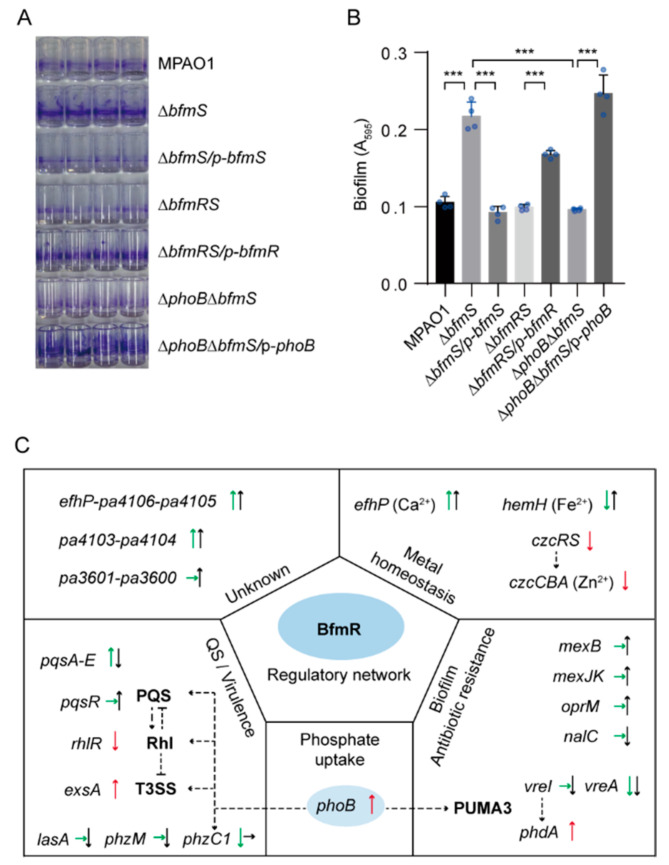
Biofilm formation assays and a proposed model of BfmR regulatory network. (**A**) Photograph showing ring-shape biofilms (stained by 1% crystal violet) near the air–liquid interface on the inner surface of polystyrene Stripwell^TM^ Microplate. (**B**) Quantification of biofilm formation in (**A**) by measuring the OD_595_ of the dye solubilized from stained biofilm. A_595_, absorbance at 595 nm as an indirect measure of biofilm biomass. MPAO1, Δ*bfmS*, Δ*bfmRS*, and Δ*phoB*Δ*bfmS* harbor an empty pAK1900 vector as a control; p-*bfmS*, p-*bfmR*, and p-*phoB* respectively denote pAK1900-*bfmS*, pAK1900-*bfmR*, and pAK1900-*phoB* plasmids (Appendix A). Data points are shown in blue dots, and results represent means ± SD (n = 4 biological replicates; *** *p* < 0.001, Student’s two-tailed *t*-test). (**C**) Proposal model of BfmR regulatory network. Some selected BfmR-targeted genes (operon) are shown. ChIP-seq signal was also observed for the promoters of *rhlR*, *phdA*, and *vreAIR* operon while it does not meet the threshold for defining BfmR targets (see detail in Appendix A). The arrow indicates the effect of BfmR on gene expression (up arrow indicates up-regulation, down arrow indicates down-regulation, and the right arrow indicates no observed effect). Green and black arrows indicate differential gene expression according to RNA-seq experiments at 6 h and 24 h, respectively; red arrow indicates differential gene expression according to either the promoter assays in this study (Figure 3E,F, Figure 5D and Appendix A) or the results of published data [11,18]. The dotted line shows the interaction between the players: arrow, activation; hammerheads, repression.

## Data Availability

Data available in a publicly accessible repository.

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
