# Peer review of "Genome-Wide Mapping Reveals Complex Regulatory Activities of BfmR in Pseudomonas aeruginosa"

_microorganisms, 2021, doi:10.3390/microorganisms9030485_

Round 1

Reviewer 1 Report

The manuscript investigates the role of the response regulator BfmR in gene regulation and overall virulence potential in P. aeruginosa.

Overall Comments:

Lines 124-128: it is not clear why the authors switched media from M8-glutamate minimal medium to LPM medium and 6 hours to 24 hours?  These conditions are not really comparable? The fact that there were more genes differentially regulated as shown in data set 3 as this is taken in post-log phase.

Lines 139 to 143: The authors speculate for possible reasons behind the discrepancy in BfmR regulated genes. The authors have potentially missed a reason due to the different timing of collection (growth phase) and growth conditions between studies.

A Statistics section should be added to the methods. What program and version was used to conduct tests etc.

Is there a reason for color coding of graphs in Figure 5 G and H? and not in 5F?

Lines 652-672: seems unnecessarily wordy. This information would be better suited in a small table, perhaps added into Table S2

Figure 4: increase font size. Even with zooming in, the names of promoters were difficult to read

Specific comments:

Line 133: do the authors mean MPAO1 genome and not PAO1 genome?

Line 618: adjusts should be "adjusted"

Line 620-621: collected the cells should be "cells were collected"

Line 626: an false  should be "a false"

Line 763: then dilute should be "then diluted"

Lines 763-766: grammar issues

Lines 775-777: grammar issues

Reviewer 2 Report

Review Genome-wide mapping reveals complex regulatory activities of BfmR in Pseudomonas aeruginosa

Comments to the Author

This is an interesting paper identifying the regulons of infection relevant transcription factors in the opportunistic human pathogen Pseudomonas aeruginosa.

The paper describes robust set of data, and the experiments include all the necessary controls. Unfortunately, the authors do not give a rationale why PhoB ChIP was done in duplicates but CzcR in triplicates. Furthermore, RNA-seq experiments were done in biological duplicates, while the gold-standard would be to perform experiments in triplicates.

The quality of the sequencing data cannot be accessed as accession "GSE154264" is currently private and is scheduled to be released on Jul 07, 2021.

The data analysis of the ChIP and RNA-seq experiments is robust and the comparisons between the different regulons is useful.

All in all, this is well-written paper. The results and data interpretation are robust.

I have some further minor points to be addressed by the authors:

TCS is not defined in the abstract

Line 37: via the two-component system (TCS) - there are more than just one TCS, it should read via two-component systems (TCS)

47-49: What does BfmR stand for?

72: What was the rationale to use these conditions? Please explain

114: ChIP, instead of CHIP (line 135, 276, 370, 403 as well)

115: 800 bp upstream of the putative translation start site of a gene or presumptive operon – 800bp seems excessively large, what was the reasoning for using such a large window. How is the distribution of peak to gene start?

127: delete “there are“

131: “BfmR is capable of modulating the expression of 2,150 genes” This reads as if BfmR directly regulates these genes, it should be stressed that secondary effects (BfmR regulates different regulators which in turn regulate genes) are playing a large part in the BfmR network

160 “because the genome-wide measurement of CzcR binding in P. aeruginosa has not been available.” can be omitted

163: coimmumoprecipitation = coimmunoprecipitation

197: I’d suggest adding negatively controlled (or something similar) to stress that BfmR inhibits expression of czcCBA/RS otherwise this sentence is a little confusing

317: remove ‘the’ before PhoB

318: remove ‘the’ before phosphate

319: remove ‘to’

340: ‘excepted’ should be expected

407: have reported

415: remove ‘the’ before PhoB

Lines 761-777 need to be re-written in past tense

Reviewer 3 Report

In this MS, Fan et al investigate the regulon of BfmR, a response regulator of two-component BfmRS in P. aeruginosa. They use ChIP-sequencing for genome-wide mapping of BfmR binding sites and identify 174 putative target regions of BfmR. Comparison of the ChIP-seq and RNA-seq data lead authors to conclude that 172 genes in 106 predicted transcription units could be regulated by BfmR transcriptional regulator. Using DNA footprinting, EMSA, reporter gene fusions, and Western blot analysis they verify that BfmR indeed binds and regulates several promoter regions including those of phzA1, czcR, exsA, phoB, oprP, oprO. Interestingly, data suggests a remarkable overlap between the BfmR, PhoB, and CzcR regulons, as multiple BfmR-regulated genes belong to PhoB or CzcR regulons. Finally, the authors propose a model of BfmR regulatory network describing how the BfmR-regulated genes are associated with several virulence factors including biofilm formation, antibiotic resistance, quorum sensing, phosphate uptake, and metal homeostasis.

I think that this manuscript presents a lot of new knowledge about BfmR regulon. The data is convincing and I have only minor concerns and suggestions. As they are mostly related to the English language and style, I would suggest text proofreading by a native English speaker (if possible).   

  1. Line 48: Replace “RR BfmR, get its name from” with “RR BfmR. The system got its name from”
  2. Line 114: Replace “Among the 161 BfmR CHIP peaks that located at intergenic regions, 160 out of them” with “160 out of 161 BfmR CHIP peaks that were found in intergenic regions”
  3. Lines 163: Replace “coimmumoprecipitation” with “coimmunoprecipitation”
  4. Figure S2A legend (line 23): Replace “analysis of BfmR binding” with “analysis of CzcR binding”
  5. Line 216, Figure 3 legend: Are you sure that the MM medium was supplemented with 50 mM ZnCl2? Something should be wrong there because such a high concentration of zinc should totally inhibit bacterial growth.
  6. Line 229: Replace “BfmR does bind to a DNA sequence covering 366 bp upstream of” with “BfmR binds to a DNA sequence locating 366 bp upstream of”
  7. Lines 232-233: Replace “and then measured its activity in aeruginosa strains including the wild-type MPAO1” with “and measured its activity in P. aeruginosa wild-type MPAO1”
  8. Lines 244-245: Replace “Red circle showing gene (or the first gene in an operon) whose promoter was co-targeted by BfmR and PhoB; yellow circle, co-targeted by…” with “Red circle denotes gene (or the first gene in an operon) whose promoter is co-regulated by BfmR and PhoB; yellow circle, gene co-regulated by…” etc
  9. Lines 269-270: Replace “(Fig. 5B, C; Fig. S4B-G)” with “(Fig. 5B, C; Fig. S5B-G)”
  10. Lines 270-271: Replace “we observed that 66 out of the 180 genes belong to the PhoB regulon exhibited” with “we observed that 66 out of the 180 PhoB regulon genes exhibited”
  11. Lines 302-304: Replace “Complementation of the ΔbfmS mutant with a plasmid-born bfmSbfmS/p-bfmS) restored the expression level of…” with “Complementation of the ΔbfmS mutant with a plasmid-born bfmSbfmS/p-bfmS) decreased the expression level of…”
  12. Lines 315-332 and Figure 5G: The usage of terms like “Pi removal” or “Pi depletion” seem to me unclear and even confusing. As the phosphate depletion assay measures the residual phosphate level in the medium, this assay indirectly measures the “phosphate uptake” or “phosphate utilization“ of the bacteria. Therefore, I would rather suggest these latter terms as biologically more relevant. For example, I would replace “BfmR boosts Pi removal and” with “BfmR boosts Pi uptake and” (line 315); “exhibited Pi depletion similar to” with “exhibited the residual Pi levels similar to” (line 322).
  13. Line 333: Replace “PhoB is required for BfmR to promote biofilm formation” with “PhoB is required for BfmR-dependent biofilm formation”
  14. Line 346: Replace “phoB plays an essential role in BfmR-biofilm formation” with “phoB plays an essential role in BfmR-dependent biofilm formation”
  15. Lines 374-376: This sentence needs to be rephrased.
  16. Line 413: Replace “it has been showed” with “it has been shown”
  17. Lines 533, 565, and 571: Replace “and then the diluted cultures were cultured in a 250 ml” with “and grown in a 250 ml”
  18. Line 538: Specify the meaning of “30” in the phrase “cells were collected from 30 OD600 of culture.
  19. Line 546: Replace “supernatant was transfer to” with “supernatant was transferred to”
  20. Line 573: Replace “After culturing for 6 hours, then performed the ChIP experiment” with “After culturing for 6 hours, the ChIP experiment was performed”
  21. Line 591: Replace “libraries prepared using” with “libraries were prepared using”
  22. Lines 620-621: Replace “After 24 h of culturing, collected the cells.” with “After 24 h incubation, the cells were collected.”
  23. Line 678: Delete double “DNA”
  24. Lines 685-686: Correct volumes (ml->μl?)
  25. Lines 760-777: The MatMet section 4.14 is confusing and should be totally rewritten.
